# Metallo-Supramolecular Hydrogels from the Copolymers of Acrylic Acid and 4-(2,2′:6′,2″-terpyridin-4′-yl)styrene

**DOI:** 10.3390/polym11071152

**Published:** 2019-07-05

**Authors:** Huiqin Zhang, Pan Liu, Zheng Chi, Xuegang Chen

**Affiliations:** Key Laboratory of Rubber-Plastic of Ministry of Education (QUST), Shandong Provincial Key Laborator of Rubber-plastics, School of Polymer Science and Engineering, Qingdao University of Science and Technology, Qingdao 266042, China

**Keywords:** terpyridine, hydrophilic polymer, metallo-supramolecular, hydrogels

## Abstract

Hydrophilic copolymers containing 2,2′:6′,2″-terpyridine moieties and acrylic acid (AA) units poly (acrylic acid-co-4-(2,2′:6′,2″-terpyridin-4′-yl)styrene) (P(AA-co-TPY)) were synthesized and characterized. Coordinated with different transition metal ions, the dilute aqueous solution of the copolymers exhibited red-shifted UV-vis absorption peaks of π-π* transition from 317 to 340 nm. Further, interacting with iron ions, the copolymer showed new absorption peaks at a longer wavelength region (570 nm) and the absorption intensity enhanced with increase of the ion concentration. When enough ions were added to coordinate with the 2,2′:6′,2″-terpyridine moieties, novel metallo-supramolecular hydrogels were obtained due to the formation of metal coordination bonds between polymer back bones and transition metal ions (Ni^2+^, Zn^2+^, Cd^2+^, Fe^2+^ and Cu^2+^), which acted as self-assembly crosslinking structures. The mechanical strength and morphology of the resulting metallo-supramolecular hydrogels have been investigated.

## 1. Introduction

While molecular chemistry is based on covalent bonds, supramolecular chemistry has been developed as the chemistry of the entities generated via intermolecular non-covalent interactions [1]. Supramolecular polymers (SP) are those in which the monomers are held together by noncovalent interactions such as host-quest recognition, hydrogen bonds, electrostatic force, metal ion coordination, ð-ð interactions and so on, and can be used to bring monomeric building blocks together to form SP backbones [2,3,4,5,6]. 

As polymers with lightly crosslinked network structures, hydrogels have been extensively studied since the 1950s [7]. Due to their unique properties such as high water content and excellent biocompatibility, hydrogels are widely used in emerging industries such as tissue engineering, biomaterials, drug releases and optoelectronic information materials [8,9,10,11,12]. Supramolecular gels are 3D networks in which the polymer main chains are crosslinked via non-covalent bonds [13,14,15,16,17,18]. Over the past few decades, enormous research interest has been directed towards the development of novel supramolecular hydrogels, in which the supramolecular crosslinking of polymer chains in water by specific, directional and dynamic non-covalent interactions occurs [19]. Due to their dynamic behavior, these novel polymeric networks usually exhibit attractive properties such as stimuli-responsiveness and self-healing for use in a wide variety of emerging applications [20]. 

Metallo-supramolecular hydrogels are gels in which the back bones are crosslinked by metal-ligand coordination. Generally, the introduction of metal ions into polymer networks usually brings a series of new physical and chemical characteristics to the gels such as photoelectric, catalytic and magnetic properties [21,22,23,24]. Liuyan Tang et al. obtained metallo-supramolecular hydrogels, based on amphiphilic polymers, with multi-stimuli responsive properties such as pH, competing ligands and reversible redox gel-sol transition with Fe^2+^ [25]. As some of the most extensively used ligands to complex transition metal ions as well as rare earth metal ions, 2,2′:6′,2″-terpyridine moiety-based ligands play an important role in design and synthesis of various metal complex-based supramolecular polymers and aggregates, which are commonly used in drug molecules, optical materials, magnetic materials and artificial enzymes, among others [26,27,28]. The easy modification of their physical properties by varying the metal ions or modifying the ligands makes them a very promising prospect [29]. Andre Duerrbeck et al. synthesized supramolecular metallo-networks in a solid state between Eu(III) and a rigid ditopic tridentate terpyridine ligand, and considered that their micron-sized fibers could be used as an efficient optical wave-guide for Eu(III) emission [30].

Herein, the preparation of novel metallo-supramolecular hydrogels, in which the complex bonds between 2,2′:6′,2″-terpyridine and transition metal ions act as the crosslinking structures of polymer networks, are reported for the first time. The key monomer (2,2′:6′,2″-terpyridin-4′-yl)styrene (TPY) was synthesized by introducing a double bond into the 2,2′:6′,2″-terpyridine-based molecule. The water-soluble polymer poly (acrylic acid-co-4-(2,2′:6′,2″-terpyridin-4′-yl)styrene (P(AA-co-TPY)), in which the fascinating ligand terpyridine moieties distributed in main chains, was obtained via radical copolymerization of acrylic acid (AA) with 4-(2,2′:6′,2″-terpyridin-4′-yl)styrene. Under induction of the transition metal ions, the polymer ligands coordinated with the transition metal ions and self-assembly crosslinked into 3D networks. Due to an abundance of hydrophilic structures (acrylic acid units) in the main chains of the resulting polymers, the coordination-crosslinked hydrogels were obtained. This work was an attempt to prepare a novel hydrogel via polymer molecular self-assembly crosslinking based on the metal-ligand coordination. 

## 2. Materials and Methods 

The monomer acrylic acid (AA) was purchased from Shanghai Huayi Acrylic Ltd (Shanghai, China) and purified by distillation under reduced pressure before use. The hydrochloride salts (for Ni^2+^, Zn^2+^, Cd^2+^, Fe^2+^, and Cu^2+^) were received from Aladdin Ltd (Shanghai, China). 1,4-dioxane were dried over sodium and refluxed. All other reagents and solvents were analytical grade. 4-(2,2′:6′,2″-terpyridin-4′-yl)toluene **1** was prepared as previously reported literature procedure [31]. 

^1^H NMR spectra of the products were determined using a BRUKER AVANCE 500 nuclear magnetic resonance spectrometer with tetramethylsilane (TMS) as the internal standard and DMSO-d_6_ or CDCl_3_ as a solvent. IR spectra was obtained by Bruker VERTEX 70 TGA-IR (Bruker, Rheinstetten, Germany). ultraviolet-visible absorption spectroscopy and measured with a HITACHI U-4100 UV-visible spectrophotometer (Hitachi, Tokyo, Japan). The rheological behaviors of the gels were determined by an ARES-G2 rheometer with parallel plate geometry of a 50-mm diameter at 25 °C. Strain sweep experiments were processed at a frequency of 1 rad/s with strain ranging from 0.1 to 1000%. Frequency sweep experiments were processed at 1% strain with frequency ranging from 0.6 to 100 rad/s. The mechanical properties of the hydrogels, mainly for compression performance, were determined by an AI-7000S Electronic tensile machine with a compression speed of 5 mm/min and the compression test samples were Φ15 × 10 mm cylinders. A JSM-7500F Scanning Electron Microscope (JEOL, Tokyo, Japan) was used to analyze the morphology of the gels. 

### 2.1. Synthesis of 4-(2,2′:6′,2″-terpyridin-4′-yl)styrene (TPY)

The terpyridine-based monomer (TPY) was prepared according to modified synthetic methods reported in the literature (Scheme 1) [31].

### 2.2. Synthesis of 4-(2,2′:6′,2″-terpyridin-4′-yl)benzyl Bromide (2)

Benzoyl peroxide (0.54 g) was added to a mixture of compound 1 (12.57 g, 38.92 mmol) and N-bromosuccinimide (6.93 g, 38.92 mmol) in carbon tetrachloride (300 mL) under nitrogen atmosphere and refluxed for 4 h. After evaporating, the solid was washed with water, dried and then recrystallized from chloroform and methanol to obtain a pale yellow crystal (12.63 g, 80.6%). ^1^H NMR (500 MH_Z_, CDCl_3_): δ (ppm) = 8.73 (s, 4 H), 8.67 (d, J = 8.0 H_Z_, 2 H), 7.88 (m, 4 H), 7.54 (d, J = 8.0 H_Z_, 2 H), 7.36 (t, J = 5.0 H_Z_, 2 H), 4.57 (s, 2 H).

### 2.3. Synthesis of 4-(2,2′:6′,2″-terpyridin-4′-yl)benzyl triphenylphosphonium Bromide (3)

The suspension of compound 2 (12.63 g, 31.38 mmol) and triphenylphosphine (8.21 g, 31.38 mmol) in dried acetone (420 mL) was refluxed for 4 h. The resulting suspension was cooled and then filtered and dried to obtain a pale yellow powder (15.66 g, 75.0%). ^1^H NMR (500 MH_Z_, CDCl_3_): δ (ppm) = 8.67 (d, J = 4.0 Hz, 2 H), 8.58 (d, J = 7.5 H_Z_, 2 H), 8.55 (s, 2 H), 7.8 (m, 11 H), 7.67 (m, 6 H), 7.61 (d, J = 8.0 H_Z_, 2 H), 7.32 (t, J = 5.0 H_Z_, 2 H), 7.25 (d, J = 6.0 H_Z_, 2 H), 5.64 (d, J = 7.5 H_Z_, 2 H).

### 2.4. Synthesis of 4-(2,2′:6′,2″-terpyridin-4′-yl)styrene (TPY)

A 10% sodium hydroxide solution was slowly added dropwise to the suspension of compound 3 and 37% formaldehyde solution. The resulting suspension was stirred for 3 h at room temperature. Then the crude product was filtered and washed several times with water. After recrystallizing in dichloromethane and methanol, pale yellow crystals (6.71 g, 84.5%) were obtained. ^1^H NMR (500 MH_Z_, CDCl_3_): δ (ppm) = 8.74 (d, J = 7.0 Hz, 4 H), 8.68 (d, J = 8.0 H_Z_, 2 H), 7.88 (m, 4 H), 7.54 (d, J = 8.0 H_Z_, 2 H), 7.35 (t, J = 6.0 H_Z_, 2 H), 6.78 (m, 1 H), 5.87 (d, J = 8.0 H_Z_, 1 H), 5.34 (d, J = 5.5 H_Z_, 1 H).

### 2.5. Synthesis of Poly (acrylic acid-co-4-(2,2′:6′,2″-terpyridin-4′-yl)styrene) (P(AA-co-TPY)) 

2,2′-Azobis(isobutyronitrile) (AIBN) (0.056 g, 0.34 mmol) was added to a mixture of TPY (1.20 g, 3.57 mmol) and acrylic acid (AA) (6.80 g, 94.36 mmol) in 7 mL 1,4-dioxane. The reaction mixture was stirred for 6 h at 80 °C. After precipitating from methylene chloride, a white solid of the resulting copolymer was obtained (5.03 g, 62.8%).

### 2.6. Preparation of the Metallo-Supramolecular Hydrogels

The polymer P(AA-co-TPY) was dissolved in water, stirred and neutralized with a 10 wt% sodium hydroxide aqueous solution; the contents of the copolymer were 6 or 8 wt%. Next, the solution of different transition metal ions—iron, zinc, nickel, copper and cadmium (0.01 mol/L)—were added into the polymer solution to obtain metal ion-induced supramolecular hydrogels when the molar ratio of cation to ligand reached 0.2–1.0/1.

## 3. Results and Discussion

### 3.1. Synthesis and Characterization of the Copolymer Ligands

The terpyridine-based monomer was prepared from the precursor 4-(2,2′:6′,2″-terpyridin-4′-yl) methylbenzene (compound 1) with a good yield. The methyl group in compound 1 firstly was bromomethylated by brominating agent N-bromosuccinimide (NBS) to give 4-(2,2′:6′,2″-terpyridin-4′-yl)benzyl bromide **2**. The bromomethyl in compound **2** reacted with triphenylphosphine to form phosphonium salt. Under the condition of a strong base, the phosphonium salt reacted with formaldehyde and the double-bond structure was formed in compound TPY. The polymerization activity of the double bond in TPY molecule was likely similar to that of styrene, so the terpyridine-based monomer TPY could copolymerize with acrylic acid initiated by the free radical initiator AIBN in 1,4-dioxane (Scheme 2). Because the resulting copolymer ligand P(AA-co-TPY) was insoluble in 1,4-dioxane, the copolymers precipitated from 1,4-dioxane after polymerization.

The successful synthesis of the copolymer P(AA-co-TPY) was proven by ^1^H NMR spectra. As seen in Figure 1, the chemical shift of the carboxyl proton in acrylic acid is approximately 12.2 ppm. The peaks of the protons in TPY units were not obvious in the spectrum due to the low content of the TPY unit in the back bone of the resulting polymer. After amplifying the spectrum (Figure 1 (inset)), the chemical shift of the protons from the terpydine-based monomer units (TPY) can be observed at 7.0–9.0 ppm can be seen. 

Infrared spectroscopy investigation was performed to further confirm the structures of the resulting copolymer. As shown in Figure 2, the stretching vibration peak of O-H in the carboxyl group (at about 3100 cm^−1^), the stretching vibration peak of C-H in -CH_2_- from polyacrylic acid (at 2932 cm^−1^) and the stretching vibration peaks of C=O (at 1711 cm^−1^) and C-O (at 1167 cm^−1^) can be well attributed. The stretching vibration peaks of the benzene ring skeleton (C=C) were at 1592 and 1534 cm^−1^, and the stretching vibration peak of the bond C-N from the pyridine heterocycles was at about 890 cm^−1^, which further proved that the copolymer containing acrylic acid and terpydine-based moiety (TPY) was synthesized successfully. 

### 3.2. UV-Visible Properties of the Dilute Copolymer Solution and the Coordination Interaction with Transition Ions

Thanks to the acrylic acid hydrophilic units, which were the main constitution of the back bone structures, the copolymer P(AA-co-TPY) had good solubility in water. The polymer was neutralized with a 10 wt% of sodium hydroxide solution and diluted to a solution with 0.05 wt% concentration. The UV-visible properties of the polymer ligand and the influences resulting from interaction of the different transition metal ions with the polymer ligand were investigated, and the results are described in Figure 3. The polymer ligand P(AA-co-TPY) exhibited a strong absorption peak at 305 nm, which was caused by the π-π* transition of the conjugated system in the 4-(2,2′:6′,2″-terpyridin-4′-yl)styrene unit. 2,2′:6′,2″-terpyridine can coordinate with many transition metal ions [26,27,28], and adding the transition metal ion solution to the diluted polymer ligand solution consequentially influenced its photophysical properties. As shown in Figure 3, after interacting with different transition metal ions, the polymer ligand exhibited red-shifts at different degrees, and the absorption peaks of the polymer ligand in the diluted aqueous solution were 313, 317, 318, 327 and 340 nm for Cu^2+^, Fe^2+^, Ni^2+^, Cd^2+^ and Zn^2+^, respectively. Compared with that of the copolymer solution without metal ion coordination (305 nm), the observed red-shifts of the absorption peaks may have resulted from the coordination of the 2,2′:6′,2″-terpyridine moiety with different transition metal ions, resulting in a change to the π-electron distribution of the conjugation structures in the polymer ligand. The degree of conjugation was enhanced, the electron delocalization in the system became greater, and the energy difference between the excited state and the ground state reduced so that the absorption peaks of the π-π* transition red-shifted after coordination of the terpyridine moieties with the transition metal ions. For Fe^2+^ in particular, there was another new absorption peak at 570 nm, which was attributed to the metal-to-ligand charge transfer (MLCT) excited state.

An investigation was made into the absorption properties of the polymer P(AA-co-TPY) aqueous solution, into which different contents of Fe^2+^ were added. As depicted in Figure 4a and Table 1, with an increase in Fe^2+^ content, both of the two absorption bands of the polymer solution increased. Further, the π-π* transition absorption peaks exhibited a slight red-shift at lower wavelengths, while the absorption peak from the MLCT transition remained invariable at longer wavelengths. As seen in Table 1, when the Fe^2+^ concentration increased from 0 to 1.0 ppm, the absorption peak from the π-π* transition red-shifted from 305 to 326 nm, and that absorption intensity is linear with the ion concentration in the visible region (Figure 4a). Further, the absorption intensity of π-π* transition peak increased from 29,790 to 62,280 L·mol^−1^·cm^−1^, and the absorption intensity from MLCT transition increased from 0 to 24,140 L·mol^−1^·cm^−1^. As seen in Figure 4b, the color of the polymer complexes with iron ions became deeper and deeper when the Fe^2+^ concentration increased, which was in accordance with the increased intensity of the absorption of 570 nm. Coordinated with other transition metal ions such as Ni^2+^, Cu^2+^, Zn^2+^ and Cd^2+^, the polymer ligand exhibited a similar trend; that is, with an increase in ion concentration, the absorption intensity from the π-π* transition increased to a different extent. Unlike Fe^2+^, however, no other new absorption band appeared.

### 3.3. Preparation and Characterization of Metallo-Supramolecular Hydrogel 

The mixing of the polymer P(AA-co-TPY) solution (after neutralization) and various divalent metal ions such as Cu^2+^, Cd^2+^, Fe^2+^, Ni^2+^ and Zn^2+^ led to gelation (the metal ions to terpyridine units ratio was 0.2–1.0:1.0), which is shown in Scheme 3 and Figure 5. The resulting hydrogels exhibited different colors based on the types of metal ions. The complex constant between the ligand and Ni^2+^ was larger [32], meaning that the formed Ni-gel was fine and the hydrogel strength was greater. The rheological properties of the Ni-gels were analyzed by rheological tests. The complex viscosities of Ni-gel (8 wt%, Figure 6a) as a function of frequency were firstly measured. As seen in Figure 6a, the complex viscosity decreased significantly with an increase of frequency, indicating that the resulting hydrogel had obvious shear thinning. Nonlinear behavior was probed using a frequency of 1 rad/s with strain ranging from 0.1 to 1000% (Figure 6b). Below 30% strain, the storage modulus (G’) and loss modulus (G’’) remained constant. Above 30% strain, the G’ value was observed to decrease rapidly, suggesting that the network of the gels had collapsed. With an increase of strain, the G’’ value increased slightly at first and then decreased. The increase of G’’ value was due to an increase in the effective volume of Ni-gel, which resulted from the extension of the polymer chain with the increase of the strain. When the G’ and G’’ were finally the same, the gel network was completely destroyed and the gel transformed into a liquid state. 

The linear response was determined by varying the frequency at 1% strain in the experiment. As shown in Figure 6c, the storage and loss modulus were independent from the applied strain. The G’ value increased slightly while the G’’ value gradually increased noticeably with increasing frequency. In addition, the G’ value was higher than the G’’ value over the whole investigated frequency scale, which identified the viscoelastic nature of Ni-gel. The solid content of the gel had a great influence on the rheological behavior of the gel. The G’ and G’’ values of the sample with 8 wt% were higher than that of the sample with 6 wt%.

The mechanical properties of the Ni-gels were also characterized by compression tests. The effects of different solid content and ion addition (or the ratio of ions/ligand) on gel properties were investigated and the results are exhibited in Figure 7. With Ni^2+^ to ligand ratios from 0.2:1 to 1:1 (the solid content was 6 wt%), the crosslinking degree of the network increased, which led to an increase in the strength of the gel. On the other hand, comparison of the compression behaviors of the different solid content gels (6 and 8 wt%) showed that the Ni-gel with larger solid content exhibited greater strength at an equal ion concentration. The decrease in strength and the increase in compression ratio were caused by the increase in water content.

In order to investigate the microstructures of the resulting metallo-supramolecular hydrogel, scanning electron microscopy (SEM) was used to characterize the metallo-supramolecular hydrogels. The gels were completely freeze-dried and then quenched, and the section was sprayed with gold to perform SEM (Figure 8). The ion species and concentrations had influences on the morphology of the formed gel. Taking Ni^2+^ as an example, the low concentration of ions (Ni^2+^:ligand = 0.2:1) added into the polymer ligand solution caused a lower crosslinking degree, and the xerogel networks were closer to two-dimensions and spread layer by layer (Figure 8a,b). With an increase in the concentration of the ions, the crosslinking degree increased gradually. When the Ni^2+^ to ligand ratio = 1:1 (Figure 8c), a three-dimensional network structure of the gel was formed and presented a tight stack structure. At the same time, although there were some differences between the Ni-gel and other gels, the overall three-dimensional network structures of the Zn-gel were found to be relatively uniform.

## 4. Conclusions

A hydrophilic copolymer containing 2,2′:6′,2″-terpyridine moieties was synthesized and well characterized. Coordinated with transition metal ions such as Ni^2+^, Zn^2+^, Cd^2+^, Fe^2+^ and Cu^2+^, the copolymers showed red-shifted absorption bands of π-π* transition in water solution. Using the appropriate ratio of copolymer ligand and metal ions, novel metallo-supramolecular hydrogels were prepared that were self-assembly crosslinked by coordination bonds between the 2,2′:6′,2″-terpyridine moieties and transition metal ions. The ion species and their concentrations had a major impact on gel formation, morphology and performances. The novel metallo-supramolecular hydrogels are promising for application in fields such as tissue engineering, catalysts, drug release, intelligent materials and so on. Further studies should detail the influences on the formation, behavior and morphology of these hydrogels, as well as the comprehensive properties carried out in the process.

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
