# Peer review of "Metallo-Supramolecular Hydrogels from the Copolymers of Acrylic Acid and 4-(2,2′:6′,2″-terpyridin-4′-yl)styrene"

_polymers, 2019, doi:10.3390/polym11071152_

Reviewer 1 Report

Review attached as a PDF file.

Author Response

The manuscript deals with the crosslinking the polymers with the use of metal cation and terpyridine unit. IN general the manuscript is well written and the text is easy to follow but I have some comments.

1. While the whole text is focused on properties of polymers that can exist in conformational equilibrium it is strongly suggested to include references to

a. publications on supramolecular polymerization,

b. publications focused on the effects of conformational preferences in supramolecular complexes.

This is suggested to make a reader more familiar with the general view of the topic and to increase the scholarly value of the manuscript.

The introduction was reorganized and added some references, which include some publications on supramoleccular polymerization and supramolecular complexes. The changed parts and words has been highlighted with yellow color.

2. The part of text in 3.1. telling about the way of synthesis is not necessary. That was already described elsewhere.

 The detailed description for synthesis is probably necessary, and the structure elucidation is main part in fact.

3. I guess the NaCl was removed from the polymer after crosslinking?

The NaCl was not removed from the polymer after crosslinking. It is difficulty to remove the NaCl from croosslingking hydrogels completely, because the Na+ and Cl+ dispersed in the hydrogel which has a certain strength. Dialysis with enough time, we think, probably can remove most of the NaCl and maybe increase water content in the hydrogels. In here, however, the preparation and the basic properties of metallo-supramolecular were focused and the existence of the NaCl has slightly influence on the formation and properties of the resulting hydrogels. For all that, the reviewer’s opinion has gave us some important enlightenment for future research.

4. While various cations were used and the polymer contains two groups that can bind cations (terpyridine and carboxylate) it is necessary to check if the cations are not, to some extent, bonded to carboxylate.

The review’s advise is very important. Although the carboxylate can bind with some cations, generally, however, this binding is not very strong. Compared with carboxylate, the terpyridine with multiple nitrogen heterocyclic rings has large enough binding competitiveness. In fact, when there is no terpydine units in main chains, the homopolymer of acrylate can not form gels after adding some transition metal cations, which confirmed the above opinion. Some binding bond with carboxylate, however, can not be excluded in this system.

Reviewer 2 Report

This paper reports the synthesis and characterization of metallo-supramolecular hydrogels based on polymeric copolymers. Some issues need to be addressed before publication:

The introduction is poor and should be improved. The Methods section should also be improved, in particular, at which concentration was the polymer P(AA-co-TPY) dissolved in water? And what were the amounts of metal ions added for the formation of the hydrogels? What polymer/metal ratios were used by the authors?

The authors state that “The complex constant between the ligand and Ni2+ is larger”. How were such constants calculated ? Please include the values in the results.

UV-visible absorption spectra of the copolymer P(AA-co-TPY) with different metal ions are reported. The concentrations used are 5 ppm for all metal ions, except for Fe2+, for which a much lower concentration was used (0.5 ppm). The authors should explain such choice and include data acquired using a concentration of  Fe2+ of 5 ppm. Also, why did the authors study only the rheological properties of Ni-based gels? Why did they not choose Fe-based gels instead? Moreover, the values of G’ recorded for Ni-based gels are very low, how could the mechanical properties of these gels be enhanced?

An accurate English revision is strongly suggested.

Author Response

1. This paper reports the synthesis and characterization of metallo-supramolecular hydrogels based on polymeric copolymers. Some issues need to be addressed before publication:

The introduction is poor and should be improved. The Methods section should also be improved, in particular, at which concentration was the polymer P(AA-co-TPY) dissolved in water? And what were the amounts of metal ions added for the formation of the hydrogels? What polymer/metal ratios were used by the authors?

The introduction was reorganized and added some references, which include some publications on supramoleccular polymerization and supramolecular complexes. The methods section has also been revised and give the necessary detailed experimental information. The polymer dissolved in water was 6 wt% or 8 wt% and the metal/ligand ratios were 0.2~1/1.  All changed parts and words has been highlighted with yellow color.

2. The authors state that “The complex constant between the ligand and Ni2+ is larger”. How were such constants calculated ? Please include the values in the results.

The expression “The complex constant between the ligand and Ni2+ is larger” is not originated from our calculation, and the conclusion come from the research result of the document. So we added a reference 32 for revised manuscript (as following).

32 Holyer, R. H.; Hubbard, C. D.; Kettle, S. F.; Wilkins, A. R. G. The kinetics of replacement reactions of complexes of the transition metals with 2,2',2"-terpyridine, Inorganic Chemistry 1966, 5, 622-625.

3. UV-visible absorption spectra of the copolymer P(AA-co-TPY) with different metal ions are reported. The concentrations used are 5 ppm for all metal ions, except for Fe2+, for which a much lower concentration was used (0.5 ppm). The authors should explain such choice and include data acquired using a concentration of  Fe2+ of 5 ppm. Also, why did the authors study only the rheological properties of Ni-based gels? Why did they not choose Fe-based gels instead? Moreover, the values of G’ recorded for Ni-based gels are very low, how could the mechanical properties of these gels be enhanced?

From the appearance, the copolymer with Fe2+ showed very deep purple and transparence was very weak. Compared with other metal ions, the molar absorptivity of the copolymer P(AA-co-TPY) with Fe2+ was rather larger. As seen in Figure 3, the molar absorptivity of 43810 L·mol-1·cm-1 for Fe2+ with content of 0.5 ppm was obtained, which was lager than that of all other metal cations. In fact, in same content (5ppm) and same instrument setting, the molar absorptivity of the Fe2+ with copolymer exceed the range of measurement and the curve exhibited obvious molecular aggregation. Based on this, the 0.5 ppm of Fe2+ content was selected to compare with other metal cations in experimental.

Likewise, the color of the Fe-based hydrogels was very deep, and the gel strength was lower than Ni-based gels. So the Ni-hydrogel was selected to study its rheological properties. For all that, as pointed out by the reviewer, the values of G’ are still low, which is the same characteristic for these metallo-supraolecular crosslinked hydrogels because that the coordination interaction between the terpyridine and transition metal ions is not very strong. Presently, we think, mechanical properties of these gels can be enhanced through increase the number of coordination bonds or increase net work evenness of the resulting metallo-hydrogel.

4. An accurate English revision is strongly suggested.

The manuscript has been checked for English and the revision parts or sentences have been highlighted.

Round  2

Reviewer 2 Report

the authors have improved the manuscript and its presentation and answered to all the points raised by the reviewers

Polymers EISSN 2073-4360 Published by MDPI AG, Basel, Switzerland RSS E-Mail Table of Contents Alert
Back to Top